# A Hitchhiker's Guide to Statistical Comparisons of Reinforcement Learning Algorithms

**Cédric Colas**[*]
INRIA - Flowers Team
Bordeaux, France

**Olivier Sigaud**
Sorbonne University - ISIR
Paris, France

**Pierre-Yves Oudeyer**
INRIA - Flowers team
Bordeaux, France

## Abstract

Consistently checking the statistical significance of experimental results is the first mandatory step towards reproducible science. This paper presents a hitchhiker's guide to rigorous comparisons of reinforcement learning algorithms. After introducing the concepts of statistical testing, we review the relevant statistical tests and compare them empirically in terms of false positive rate and statistical power as a function of the sample size (number of seeds) and effect size. We further investigate the robustness of these tests to violations of the most common hypotheses (normal distributions, same distributions, equal variances). Beside simulations, we compare empirical distributions obtained by running Soft-Actor Critic and Twin-Delayed Deep Deterministic Policy Gradient on Half-Cheetah. We conclude by providing guidelines and code to perform rigorous comparisons of RL algorithm performances.

## 1 Introduction

Reproducibility in Machine Learning and Reinforcement Learning in particular (RL) has become a serious issue in the recent years. As pointed out in Islam et al. [1] and Henderson et al. [2], reproducing the results of an RL paper can turn out to be much more complicated than expected. In a thorough investigation, Henderson et al. [2] showed it can be caused by differences in codebases, hyperparameters (e.g. size of the network, activation functions) or the number of random seeds used by the original study. Henderson et al. [2] states the obvious: the claim that an algorithm performs better than another should be supported by evidence, which requires the use of statistical tests. Building on these observations, this paper presents a hitchhiker's guide for statistical comparisons of RL algorithms. The performances of RL algorithm have specific characteristics (they are independent of each other, they are not paired between algorithms etc.). This paper reviews some statistical tests relevant in that context and compares them in terms of false positive rate and statistical power. Beside simulations, it compares empirical distributions obtained by running Soft-Actor Critic (SAC) [3] and Twin-Delayed DDPG (TD3) [4] on Half-Cheetah [5]. We finally provide guidelines to perform robust difference testing in the context of RL. A repository containing the raw results and the code to reproduce all experiments is available at `https://github.com/ccolas/rl_stats`.

## 2 Comparing RL Algorithms: Problem Definition

### 2.1 Model

In this paper, we consider the problem of conducting meaningful comparisons of Algorithm 1 and Algorithm 2. Because the seed of the random generator is different for each run[2], two runs of a

---

[*]cedric.colas@inria.fr

[2] Yes, it should be. The random seed is not an hyperparameter.

Preprint. Work in progress.

same algorithm yield different measures of performance. An algorithm performance can therefore be modeled as a random variable $X$, characterized by a distribution. Measuring the performance $x$ at the end of a particular run is equivalent to measuring a *realization* of that random variable. Repeating this $N$ times, we obtain a *sample* $x = (x^1, ..., x^N)$ of size $N$.

To compare RL algorithms on the basis of their performances, we focus on the comparisons of the central tendencies ($\mu_1$, $\mu_2$): the means or the medians of the associated random variables $X_1$, $X_2$.[3] Unfortunately, we cannot know $\mu_1$, $\mu_2$ exactly. Given a sample $x_i$ of $X_i$, we can estimate $\mu_i$ by the empirical mean: $\overline{x}_i = 1/N \sum_{j=1}^{N} x_i^j$ (resp. the empirical median). However, comparing central performances does not simply boil down to the comparison of their estimates. As an illustration, Figure 1 shows two normal distributions describing the distributions of two algorithm performances $X_1$ and $X_2$. Two samples of sample size $N = 3$ are collected. In this example, we have $\mu_1 < \mu_2$ but $\overline{x}_1 > \overline{x}_2$. The rest of this text uses *central*

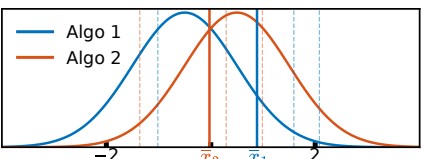

Figure 1: **Two normal distributions representing the performances of two algorithms.** Dashed lines: performance measures (realizations). Plain lines: empirical means of the two samples ($N = 3$).

*performance* to refer to either the mean or the median of the performance distribution $i$. It is noted $\mu_i$ while its empirical estimate is noted $\overline{x}_i$. The distinction is made where necessary.

## 2.2 A Few Definitions

**Statistical difference testing.** Statistical difference testing offers a principled way to compare the central performances of two algorithms. It defines two hypothesis: 1) the *null hypothesis* $\mathcal{H}_0$: $\Delta\mu = \mu_1 - \mu_2 = 0$ and 2) the *alternative hypothesis* $\mathcal{H}_a$: $|\Delta\mu| > 0$. When performing a test, one initially assumes the null hypothesis to be true. After having observed $(x_1, x_2)$, statistical tests usually estimate the probability to observe two samples whose empirical central difference is at least as extreme as the observed one ($|\Delta\overline{x}| = |\overline{x}_1 - \overline{x}_2|$) under $\mathcal{H}_0$ (e.g. given $\Delta\mu = 0$). This probability is called the *p-value*. If the p-value is very low, the test rejects $\mathcal{H}_0$ and concludes that a true underlying difference ($\mathcal{H}_a$) is likely. When the p-value is high, the test does not have enough evidence to conclude. This could be due to the lack of true difference, or to the lack of statistical power (too few measurements given how noisy they are). The significance level $\alpha$ (usually $\leq 0.05$) draws the line between rejection and conservation of $\mathcal{H}_0$: if p-value $< \alpha$, $\mathcal{H}_0$ is rejected.

**Statistical errors.** Note that having a p-value of 0.05 still results in 1 chance out of 20 to claim a difference that does not exist. This is called a *type-I error* or *false positive*. The false positive rate is usually noted $\alpha$, just like the significance level. Indeed, statistical tests with significance level $\alpha$ are supposed to enforce a false positive

Table 1: Hypothesis testing

|  | True $\mathcal{H}_0$ | True $\mathcal{H}_a$ |
|---|---|---|
| Pred. $\mathcal{H}_0$ | True neg. $1-\alpha^*$ | False neg. $\beta^*$ |
| Pred. $\mathcal{H}_a$ | False pos. $\alpha^*$ | True pos. $1-\beta^*$ |

rate of $\alpha$. Further experiments demonstrate it is not always the case, which is why we prefer to note the false positive rate $\alpha^*$. False negatives occur when the statistical test fails to recognize a true difference in the central performances. This depends on the size of the underlying difference: the larger the difference, the lower the risk of false negative. The false negative rate is noted $\beta^*$.

**Trade-off between false positive and statistical power.** Ideally, we would like to set $\alpha = 0$ to ensure the lowest possible false positive rate $\alpha^*$. However, decreasing the confidence level makes the statistical test more conservative. The test requires even bigger empirical differences $\Delta\overline{x}$ to reject $\mathcal{H}_0$, which decreases the probability of true positive. This probability of true positive $1-\beta^*$ is called the statistical power of a test. It is the probability to reject $\mathcal{H}_0$ when $\mathcal{H}_a$ holds. It is directly impacted by the effect size: the larger the effect size, the easier it is to detect (larger statistical power). It is also a direct function of the sample size: larger samples bring more evidence to support the rejection of $\mathcal{H}_0$. Generally, the sample size is chosen so as to obtain a theoretical statistical power of $1-\beta^* = 0.8$. Different tests have different statistical powers depending on the assumptions they make, whether they are met, how the p-value is derived etc.

---

[3] Because of space constraints, we do not investigate other possible criteria for comparing RL algorithms (e.g. lower variance, high minimal performance, area under the learning curve, etc.)

**Parametric vs. non-parametric.** Parametric tests usually compare the means of two distributions by making assumptions on the distributions of the two algorithms' performances. Non-parametric tests on the other hand usually compare the medians and do not require assumptions on the type of distributions. Non-parametric tests are often recommended when one wants to compare median rather than means, when the data is skewed or when the sample size is small. Section 4.2 shows that these recommendations are not always justified.

**Test statistic.** Statistical tests usually use a *test statistic*. It is a numerical quantity computed from the samples that summarizes the data. In the t-test for instance, the statistic $t_\alpha$ is computed as $t_\alpha = |\Delta \overline{x}|/\sigma_{pool}$, where $\sigma_{pool}$ is the pooled standard deviation ($\sigma_{pool} = \sqrt{(\sigma_1^2 + \sigma_2^2)/2}$). Under the t-test assumptions, this statistic follows the analytic Student's distribution with density function $f_S(t)$. The probability to obtain a difference more important than the sample difference $\Delta \overline{x}$ (p-value) can be rewritten p-value $= P(|t| > t_\alpha)$ and can be computed as the area under $f_S(t)$ such that $|t| > t_\alpha$.

**Relative effect size.** The *relative effect size* $\epsilon$ is the absolute effect size $|\Delta \mu|$, scaled by the pooled standard deviation $\sigma_{pool}$, such that $\epsilon = |\Delta \mu|/\sigma_{pool}$. The relative effect size is independent of the spread of the considered distributions.

## 3  Statistical Tests for RL

### 3.1  Assumptions in the Context of RL

Each test makes some assumptions (e.g. about the nature of the performance distributions, their variances, the sample sizes etc.). In the context of RL, some assumptions are reasonable while others are not. It is reasonable to assume that RL performances are measured at random and independently from one another. The samples are not paired, and here we assume they have the same size.[4] Other common assumptions might be discussed:

- Normal distributions of performances: this might not be the case (skewed distributions, bimodal distributions, truncated distributions).
- Continuous performances: the support of the performance distribution might be bounded: e.g. in the Fetch environments of Gym [5], the performance is a success rate in $[0, 1]$.
- Known standard deviations: this is not the case in RL.
- Equal standard deviations: this is often not the case (see [2]).

### 3.2  Relevant Statistical Tests

This section briefly presents various statistical tests relevant to the comparison of RL performances. It focuses on the underlying assumptions [6] and provides the corresponding implementation from the Python *Scipy* library when available. Further details can be found in any statistical textbook. Contrary to Henderson et al. [2], we do not recommend using the Kolmogorov-Smirnov test as it tests for the equality of the two distributions and does not test for a difference in their central tendencies [7].

**T-test.** This parametric test compares the means of two distributions and assumes the two distributions have equal variances [8]. If this variance is known, a more powerful test is available: the Z-test for two population means. The test is accurate when the two distributions are normal, it gives an approximate guide otherwise. Implementation: *scipy.stats.ttest_ind(x1, x2, equal_var=True)*.

**Welch's t-test.** It is a t-test where the assumption of equal variances is relaxed [9]. Implementation: *scipy.stats.ttest_ind(x1, x2, equal_var=False)*.

**Wilcoxon Mann-Whitney rank sum test.** This non-parametric test compares the median of two distributions. It does not make assumptions about the type of distributions but assumes they are continuous and have the same shape and spread [10]. Implementation: *scipy.stats.mannwhitneyu(x1, x2, alternative='two-sided')*.

---

[4] This assumption could be relaxed as none of the test requires it.

**Ranked t-test.** In this non-parametric test that compares the medians, all realizations are ranked together before being fed to a traditional t-test. Conover and Iman [11] shows that the computed statistic is a monotonically increasing function of the statistic computed by the Wilcoxon Mann-Whitney test, making them really close. Implemented in our code.

**Bootstrap confidence interval test.** In the bootstrap test, the sample is considered to be an approximation of the original distribution [12]. Given two observed samples $(x_1, x_2)$ of size $N$, we obtain two bootstrap samples $(\tilde{x}_1, \tilde{x}_2)$ of size $N$ by sampling with replacement in $(x_1, x_2)$ respectively and compute the difference in empirical means $\Delta\tilde{x}$. This procedure is repeated a large number of times (e.g. $10^3$). The distance between percentiles $\frac{\alpha\times100}{2}$ and $100(1-\frac{\alpha}{2})$ is considered to be the $100(1-\alpha)\%$ confidence interval around the true mean difference $\Delta\mu$. If it does not include 0, the test rejects the null hypothesis with confidence level $\alpha$. This test does not require any assumptions on the performance distributions, but we will see it requires large sample sizes. Implementation: `https://github.com/facebookincubator/bootstrapped`.

**Permutation test.** Under the null hypothesis, the realizations of both samples would come from distributions with the same mean. The empirical mean difference $(\Delta\overline{x})$ should not be affected by the relabelling of the different realization (in average). The permutation test performs permutations of the realization labels and computes $\Delta\tilde{x} = \tilde{x}_1 - \tilde{x}_2$. This procedure is repeated many times (e.g. $10^3$). $\mathcal{H}_0$ is rejected if the proportion of $|\Delta\tilde{x}|$ that falls below the original difference $|\Delta\overline{x}|$ is higher than $1-\alpha$. Implemented in our code.

## 4 Empirical Comparisons of Statistical Tests

This section compares the above statistical tests in terms of their false positive rates and statistical powers. A false positive rate estimates the probability to claim that two algorithms perform differently when $\mathcal{H}_0$ holds. It impacts directly the reproducibility of a piece of research and should be as low as possible. Statistical power is the true positive rate and refers to the probability to find evidence for an existing effect. The following study is an extension of the one performed in [13]. We conduct experiments using models of RL distributions (analytic distributions) and true empirical RL distributions collected by running 192 trials of both SAC [3] and TD3 [4] on Half-Cheetah-v2 [5] for 2M timesteps.[5]

### 4.1 Methods

**Investigating the case of non-normal distributions.** Several candidate distributions are selected to model RL performance distributions (Figure 2): a standard normal distribution, a log-normal distribution and a bimodal distribution that is an even mixture of two normal distributions. All these distributions are tuned so that $\mu = 0, \sigma = 1$. In addition we use two empirical distributions of size 192 collected from SAC and TD3.

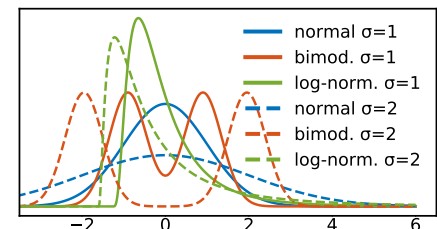

Figure 2: **Candidate distributions to represent algorithm performances**.

**Investigating the case of unequal standard deviations.** To investigate the effect of unequal standard deviations, we tune the distribution parameters to double the standard deviation of Algorithm 2 as compared to Algorithm 1. We also compare SAC and TD3 which have different standard deviations ($\sigma_{TD3} = 1.15\,\sigma_{SAC}$).

**Measuring false positive rates.** To test for false positive rates $\alpha^*$, we simply enforce $\mathcal{H}_0$ by aligning the central performances of the two distributions: $\mu_1 = \mu_2 = 0$ (the median for the Mann-Whitney test and the ranked t-test, the mean for others). Given one test, two distributions and a sample size, we sample $x_1$ and $x_2$ from distributions $X_1$, $X_2$ and compare them using the test with $\alpha = 0.05$. We repeat this procedure $N_r = 10^3$ times and estimate $\alpha^*$ as the proportion of $\mathcal{H}_0$ rejection.

---

[5] Using the spinning up implementation of OpenAI: `https://github.com/openai/spinningup`

The standard error of this estimate is: $se(\alpha^*) = \sqrt{(\alpha^*(1-\alpha^*)/N_r}$. It is smaller than the widths of the lines on the reported figures. This procedure is repeated for every test, every combination of distributions and for several sample sizes (see pseudo-code in the supplementary material).

**Measuring true positive rates (statistical power).**   Here, we enforce the alternative hypothesis $\mathcal{H}_a$ by sampling $x_1$ from a given distribution centered in 0 (mean or median depending on the test), and $x_2$ from a distribution whose mean (resp. median) is shifted by an effect size $\Delta\mu$. Given one test, two distributions (the second being shifted) and the sample size, we repeat the procedure above and obtain an estimate of the true positive rate. Tables reporting the statistical powers for various effect sizes, sample sizes, tests and assumptions are made available in the supplementary results.

## 4.2   Results: Comparison of False Positive Rates

**Same distributions, equal standard deviations.**   Figure 3(a) and 3(b) represent the false positive rates $\alpha^*$ as a function of the sample size (number of seeds), for various tests when the samples are drawn from **(a)**: the same standard normal distribution (ideal situation, all assumptions are met), and **(b)**: the same bimodal distribution. Given the sample sizes used estimate $\alpha^*$ ($N_r = 10^3$), we can directly compare the mean estimates (the lines) to the significance level $\alpha = 0.05$, the standard errors being smaller than the widths of these lines.[6] $\alpha^*$ is very large when using bootstrap tests, unless large sample sizes are used ($>40$). Using small sample sizes ($<5$), the permutation and the ranked t-test also show large $\alpha^*$. Results using two log-normal distributions show similar behaviors and can be found in the supplementary results.

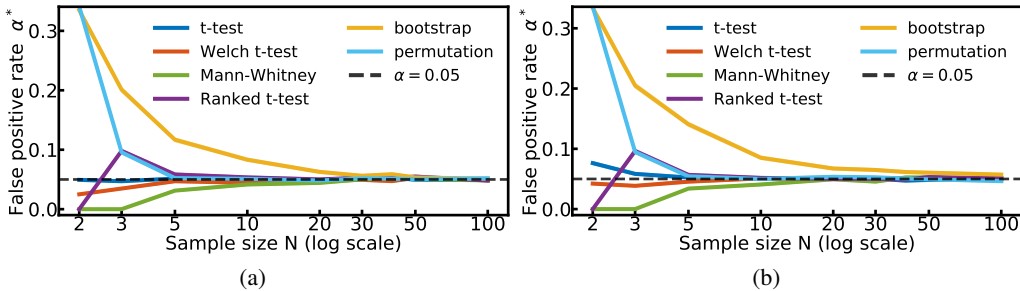

Figure 3: **False positive rates for same distributions, equal standard deviations.** Both samples are drawn from the same distribution ($\mu = 0$, $\sigma = 1$). **(a)**: A standard normal distribution. **(b)**: A bimodal distribution.

**Same distributions, unequal standard deviations.**   Here, we sample $x_1$ from a distribution, and $x_2$ from the same type of distribution with doubled standard deviation. Comparing two normal distributions with different standard deviation does not differ much from the case with equal standard deviations. Figure 4(a) (bimodal distributions) shows that Mann-Whitney and ranked t-test (median tests) constantly overestimate $\alpha^*$, no matter the sample size ($\alpha^* > 0.1$). For log-normal distributions on the other hand (Figure 4(b)), the false positive rate using these tests respects the confidence level ($\alpha^* \leq \alpha$) with sample sizes higher than $N = 10$. However, other tests tend to show large $\alpha^*$, even for large sample sizes ($\alpha^* \approx 0.07$ up to $N > 50$).

**Different distributions, equal standard deviations.**   Now we compare samples coming from different distributions with equal standard deviations. Comparing normal and bimodal distributions of equal standard deviation does not impact much the false positive rates curves (similar to Figure 3(a)). However, Figure 5(a) and 5(b) show that when one of the two distributions is skewed (log-normal), the Mann-Whitney and the ranked t-test demonstrate very important false positive rate, a phenomenon that gets worse with larger sample sizes. Section 4.5 discusses why it might be the case.

**Different distributions, unequal standard deviations.**   We now combine different distributions and different standard deviations. As before, comparing a skewed distribution (log-normal) and a symmetric one leads to high false positive rates for the Mann-Whitney test and the ranked t-test

---

[6]We reproduced all the results twice, hardly seeing any difference in the figures.

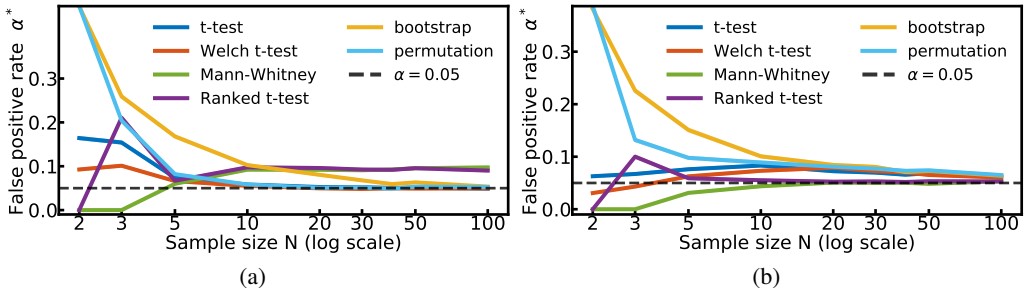

(a)            (b)

Figure 4: **False positive rates for same distributions, different standard deviations.** $x_1$ and $x_2$ are drawn from the same type of distribution, centered in 0 (mean or median), with $\sigma_1 = 1$ and $\sigma_2 = 2$. **(a)**: Two bimodal distributions. **(b)**: Two log-normal distributions.

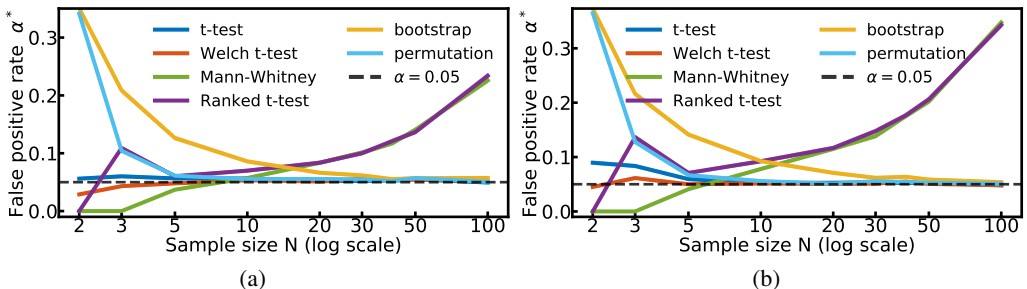

(a)            (b)

Figure 5: **False positive rates for different distributions, equal standard deviations.** $x_1$ and $x_2$ are drawn from two different distributions, centered in 0 (mean or median), with $\sigma_1 = \sigma_2 = 1$. **(a)**: normal and log-normal distributions. **(b)**: bimodal and log-normal distributions.

(Figure 6(a) and 6(b)). Comparing a normal distribution and a skewed log-normal with higher standard deviation leads to high positive rates for all other tests as well ($\alpha^* \approx 0.1$), even using large sample sizes (Figure 6(a)).

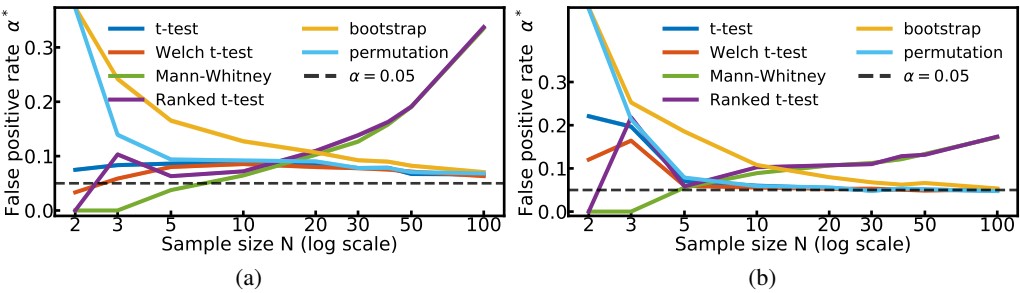

(a)            (b)

Figure 6: **False positive rates for different distributions, different standard deviations.** $x_1$ and $x_2$ are drawn from two different distributions, centered in 0 (mean or median), with $\sigma_1 = 1$ and $\sigma_2 = 2$. **(a)**: normal and log-normal distributions. **(b)**: bimodal and log-normal distributions.

## 4.3 Results: Comparison of Statistical Powers

All tests show similar estimations of statistical power. More than 50 samples are needed to detect a relative effect size $\epsilon = 0.5$ with $80\%$ probability, close to 20 with $\epsilon = 1$ and a bit more than 10 with $\epsilon = 2$. Tables reporting statistical power for all conditions, tests, sample sizes and relative effect sizes are provided in the supplementary results.

## 4.4 Results: Comparison of Real RL Distributions: SAC and TD3

Finally, we compare two empirical distributions obtained from running two RL algorithms (SAC, TD3) 192 times each, on Half-Cheetah. We observe a small increase in false positive rates when using the ranked t-test (Figure 7). The relative effect size estimated from the empirical distributions is $\epsilon = 0.80$ (median), or $\epsilon = 0.93$ (mean), in favor of SAC. For such relative effect sizes, the sample sizes required to achieve a statistical power of $0.8$ are between 10 and 15 for tests comparing the mean and between 15 and 20 for tests comparing the median (see full table in supplementary results). Using a sample size $N = 5$ with the Welch's t-test, the effect size would need to be 3 to 4 times larger to be detected with $0.8$ probability.

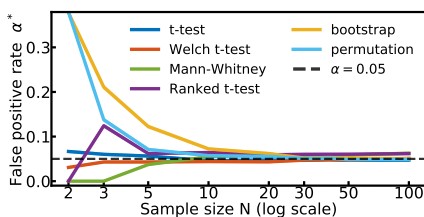

Figure 7: **False positive rates when comparing SAC and TD3.** $x_1$ is drawn from SAC performances, $x_2$ from TD3 performances. Both are centered in $0$ (mean or median), with $\sigma_1 = 1.313$ and $\sigma_2 = 1.508$.

## 4.5 Discussion of Empirical Results

**No matter the distributions.** From the above results, it seems clear that the bootstrap test should never be used for sample sizes below $N = 50$ and the permutation test should never be used for sample sizes below $N = 10$. The bootstrap test in particular, uses the sample as an estimate of the true performance distribution. A small sample is a very noisy estimate, which leads to very high false positive rates. The ranked t-test shows a false positive rate of $0$ and a statistical power of $0$ when $N = 2$ in all conditions. As noted in [13], comparing two samples of size $N = 2$ can result in only four possible p-values (only 4 possible orders when ranked), none of which falls below $\alpha = 0.05$. Such quantization issues make this test unreliable for small sample sizes, see [13] for further comments and references on this issue.

**When distributions do not meet assumptions.** In addition to the behaviors reported above, Section 4.2 shows that non-parametric tests (Mann-Whitney and ranked t-test) can demonstrate very high false positive rates when comparing a symmetric distribution with a skewed one (log-normal). This effect gets worse linearly with the sample size. When the sample size increases, the number of samples drawn in the skewed tail of the log-normal increases. All these realizations will be ranked above any realizations from the other distribution. Therefore, the larger the sample size, the more realization are ranked first in favor of the log-normal, which leads to a bias in the statistical test. This problem does not occur when two log-normal are compared to one another. Comparing a skewed distribution to a symmetric one violates the Mann-Whitney assumptions stating that distributions must have the same shape and spread. The false positive rates of Mann-Whitney and ranked t-test are also above the confidence level whenever a bimodal distribution is compared to another distribution. The traditional recommendation to use non-parametric tests when the distributions are not normal seems to be failing when the two distributions are different.

**Most robust tests.** The t-test and the Welch's t-test were found to be more robust than others to violations of their assumptions. However, $\alpha^*$ was found to be slightly above the required level ($\alpha^* > \alpha$) when at least one of the two distributions is skewed ($\alpha^* \approx 0.1$) no matter the sample size, and when one of the two distributions is bimodal, for small sample sizes $N < 10$. Welch's $\alpha^*$ is always a bit lower than the t-test's $\alpha^*$.

**Statistical power.** Except for the anomalies in small sample size mentioned above due to overconfident tests like the bootstrap or the permutation tests, statistical powers stay qualitatively stable no matter the distributions compared, or the test used: $\epsilon = 0.5$: $N \approx 100$; $\epsilon = 1$: $N \approx 20$ and $\epsilon = 2$: $N \approx 5, 10$.

## 5 Guidelines for Comparison of RL Algorithm Performances

**Measuring the performance of RL Algorithms.** Before using any statistical test, one must obtain measures of performance. RL algorithms should ideally be evaluated *offline*. The algorithm performance after $t$ steps is measured as the average of the returns over $E$ evaluation episodes conducted

independently from training, usually using a deterministic version of the current policy (e.g. $E = 20$). Evaluating agents by averaging performances over several training episodes results in a much less interpretable performance measure and should be stated clearly. The collection of performance measures forms a learning curve.

**Representing learning curves.** After obtaining a learning curve for each of the $N$ runs, it can be rendered on a plot. At each evaluation, one can represent either the empirical mean or median. Whereas the empirical median directly represents the center of the collected sample, the empirical mean tries to model the sample as coming from a Gaussian distribution, and under this assumptions represents the maximum likelihood estimate of that center. Error bars should also be added to this plot. The standard deviation (SD) represents the variability of the performances, but is only representative when the values are approximately normally distributed. When it is not normal, one should prefer to represent interpercentile ranges (e.g. $10\% - 90\%$). If the sample size is small (e.g. $<10$), the most informative solution is to represent all learning curves in addition to the mean or median. When performances are normally distributed, the standard error of the mean (SE) or confidence intervals can be used to represent estimates of the uncertainty on the mean estimate.

**Robust comparisons. Which test, which sample sizes?** The results in Section 4.2 advocate for the use of the Welch's t-test, which shows lower false positive rate and similar statistical powers than other tests. However, the false positive rate often remains superior to the confidence level $\alpha^* > \alpha$ when the distributions are not normal. When in doubt, we recommend using lower confidence levels $\alpha < 0.05$ (e.g. $\alpha = 0.01$) to ensure that $\alpha^* < 0.05$. The number of random seeds to be used to achieve a statistical power of $0.8$ depends on the expected relative effect size: $\epsilon = 0.5$: $N \approx 100$; $\epsilon = 1$: $N \approx 20$ and $\epsilon = 2$: $N \approx 5, 10$. The analysis of a real case comparing SAC and TD3 algorithms, required a sample size between $N = 10$ and $N = 15$ for a relatively strong effect $\epsilon = 0.93$ when comparing the means, and about 5 more seeds when comparing the medians ($\epsilon = 0.80$). Small sample sizes like $N = 5$ would require 3 to 4 times larger effects.

**A word on multiple comparisons.** When performing multiple comparisons (e.g. between different pairs of algorithms evaluated in the same setting), the probability to have at least one false positive increases linearly with the number of comparisons $n_c$. This probability is called the Family-Wise Error Rate (FWER). To correct for this effect, one must apply corrections. The Bonferroni correction for instance adapts the confidence level $\alpha_{Bonf.} = \alpha/n_c$ [14]. This ensures that the FWER stays below the initial $\alpha$. Using this corrections makes each test more conservative and decreases its statistical power.

**Comparing full learning curves.** Instead of only comparing the final performances of the two algorithms after $T$ timesteps in the environment, we can compare performances along learning. This consists in performing a statistical comparison for every evaluation step. This might reveal differences in speed of convergence and can provide more robust comparisons. Further discussions on how this relates to the problem of multiple comparison is given in the supplementary materials.

# 6  Conclusion

In conclusion, this paper advocates for the use of Welch's t-test with low confidence level ($\alpha < 0.05$) to ensure a false positive rate below $\alpha^* < 0.05$. The sample size must be selected carefully depending on the expected relative effect size. It also warns against the use of other unreliable tests, such as the bootstrap test (for $N < 50$), the Mann-Whitney and the ranked t-test (unless assumptions are carefully checked), or the permutation test (for $N < 10$). Using the t-test or the Welch's t-test with small sample sizes ($<5$) usually leads to high false positive rate and would require very large relative effect sizes (over $\epsilon = 2$) to show good statistical power. Sample sizes above $N = 20$ generally meet the requirement of a $0.8$ statistical power for a relative effect size $\epsilon = 1$.

**Acknowledgments**

This research is financially supported by the French Ministère des Armées - Direction Générale de l'Armement.

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

# 7 Supplementary Methods

## 7.1 Pseudo-code

Algorithm 1 represents the pseudo-code of the experiment. The whole code can be found at `https://github.com/ccolas/rl_stats`. *distributions* refers to a list of pairs of distributions. When comparing tests for an equal distribution setting, the pairs represent twice the same type of distribution. When comparing for an unequal variance setting, the standard deviation of the second distribution is doubled. The number of repetitions is set to $10.000$. The *rejection* variable refers to the rejection of the null hypothesis. The false positive error rates can be found in results_array[:, :, 0, :] when there is no shift between the distributions (null effect size), while the statistical powers are found in results_array[:, :, 1:, :].

---

**Algorithm 1** Comparisons of statistical tests

---

1: **Input:** distributions, tests, nb_repets, effect_sizes, sample_sizes, $\alpha$
2: **Initialize:** results_array           ▷ of size (nb_distrib, nb_tests, nb_effects, nb_sample_sizes)
3: **for** i_d, distrib in distributions **do**
4:      **for** i_t, test in tests **do**
5:          **for** i_e, effect_size in effect_sizes **do**
6:              **for** i_ss, N in sample_sizes **do**
7:                  rejection_list = []
8:                  **for** i_r = 1: nb_repets **do**
9:                      distrib[1].shift(effect)
10:                      sample1 = distrib[0].sample(N)
11:                      sample2 = distrib[1].sample(N)
12:                      rejection_list.append(test.test(sample1, sample2, $\alpha$))
13:              results_array[i_d, i_t, i_e, i_ss] = mean(rejection_list)

---

## 7.2 Correcting for Multiple Comparison when Comparing Learning Curves

The correction to apply when comparing two learning curves depends 1) on the number of comparisons, 2) on the criteria that is used to conclude whether an algorithm is better than the other. The criteria used to draw a conclusion must be decided before running any test. An example can be: *if when comparing the last* 100 *performance measures of the two algorithms, more than* 50 *comparisons show a significant difference, then Algorithm 1 is better than Algorithm 2*. In that case, the number of comparisons performed is $N_c = 100$, and the criterion is $N_{rejection} > N_{crit} = 50$. We want to constrain the probability FWER that our criterion is met by pure chance to a confidence level $\alpha$=0.05. This probability is: FWER $= \alpha \times N_c/N_{crit}$. To make it satisfy FWER $= 0.05$, we need to correct $\alpha$ such as $\alpha_{corrected} = \alpha \times N_{crit}/N_c$ ($\alpha_{corrected} = \alpha/2$ in our case).

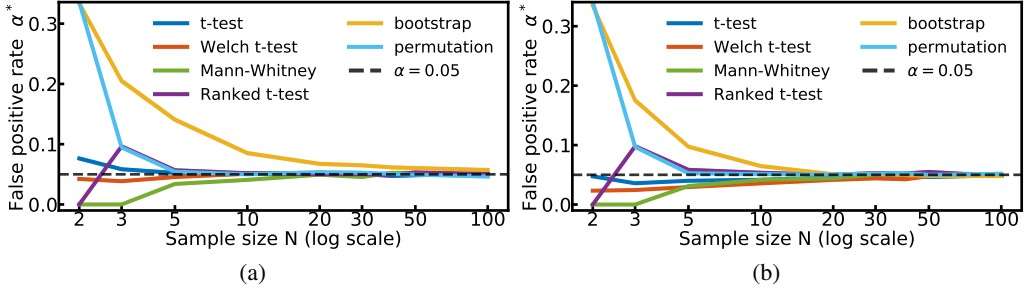

(a)                 (b)

Figure 8: **False positive rates for same distributions, equal variances.** Both samples are drawn from the same distribution. **(a)**: A bimodal distribution ($\mu = 0$, $\sigma = 1$). **(b)**: A skewed log-normal distribution ($\mu = 0$, $\sigma = 1$).

# 8 Supplementary Results

## 8.1 Comparing same distributions with equal standard deviations.

Table 2: **Statistical power when comparing samples from two normal distribution with equal standard deviation:** $(\mu_1 = 0, \sigma_1 = 1)$, $(\mu_2 = \epsilon\,\sigma_{pool}, \sigma_2 = 1)$. Each result represents the percentage of true positive over $10.000$ repetitions. In bold are results satisfying a true positive rate above $0.8$.

| N | t-test | Welch | Mann-Whit. | r. t-test | boot. | permut. |
|---|---|---|---|---|---|---|
| \multicolumn{7}{l}{Small relative effect size: $\epsilon = 0.5$} |
| 2 | 0.048 | 0.024 | 0.000 | 0.000 | 0.298 | 0.300 |
| 3 | 0.072 | 0.046 | 0.000 | 0.128 | 0.229 | 0.122 |
| 5 | 0.106 | 0.089 | 0.065 | 0.114 | 0.206 | 0.105 |
| 10 | 0.179 | 0.186 | 0.167 | 0.184 | 0.256 | 0.182 |
| 20 | 0.336 | 0.340 | 0.321 | 0.332 | 0.378 | 0.341 |
| 30 | 0.480 | 0.478 | 0.458 | 0.449 | 0.513 | 0.477 |
| 40 | 0.604 | 0.592 | 0.567 | 0.576 | 0.611 | 0.588 |
| 50 | 0.691 | 0.693 | 0.678 | 0.680 | 0.717 | 0.693 |
| 100 | **0.943** | **0.940** | **0.929** | **0.932** | **0.947** | **0.940** |
| \multicolumn{7}{l}{Medium relative effect size: $\epsilon = 1.0$} |
| 2 | 0.094 | 0.045 | 0.000 | 0.000 | 0.456 | 0.461 |
| 3 | 0.155 | 0.115 | 0.000 | 0.258 | 0.411 | 0.251 |
| 5 | 0.284 | 0.269 | 0.205 | 0.289 | 0.461 | 0.295 |
| 10 | 0.560 | 0.553 | 0.506 | 0.550 | 0.646 | 0.556 |
| 20 | **0.870** | **0.862** | **0.857** | **0.850** | **0.894** | **0.869** |
| 30 | **0.970** | **0.966** | **0.957** | **0.960** | **0.974** | **0.969** |
| \multicolumn{7}{l}{Large relative effect size: $\epsilon = 2.0$} |
| 2 | 0.217 | 0.108 | 0.000 | 0.000 | 0.773 | 0.787 |
| 3 | 0.473 | 0.370 | 0.000 | 0.626 | **0.801** | 0.593 |
| 5 | 0.788 | 0.771 | 0.675 | 0.780 | **0.914** | 0.788 |
| 10 | **0.987** | **0.988** | **0.979** | **0.984** | **0.993** | **0.990** |

Table 3: **Statistical power when comparing samples from two bimodal distribution with equal standard deviation.** The first is centered in $0$ ($\mu_1 = 0$, mean or median depending on the test), the other shifted by the relative effect size ($\mu_2 = \epsilon \, \sigma_{pool}$). Both have same standard deviation $\sigma_1 = \sigma_2 = 1$. Each result represents the percentage of true positive over $10.000$ repetitions. In bold are results satisfying a true positive rate above $0.8$.

| Small relative effect size: $\epsilon = 0.5$ | | | | | |
|---|---|---|---|---|---|
| N | t-test | Welch | Mann-Whit. | r. t-test | boot. | permut. |
| 2 | 0.064 | 0.035 | 0.000 | 0.000 | 0.291 | 0.291 |
| 3 | 0.061 | 0.041 | 0.000 | 0.122 | 0.202 | 0.119 |
| 5 | 0.091 | 0.084 | 0.075 | 0.119 | 0.193 | 0.092 |
| 10 | 0.168 | 0.168 | 0.179 | 0.198 | 0.243 | 0.174 |
| 20 | 0.325 | 0.326 | 0.362 | 0.363 | 0.367 | 0.317 |
| 30 | 0.460 | 0.469 | 0.505 | 0.509 | 0.503 | 0.456 |
| 40 | 0.592 | 0.582 | 0.632 | 0.639 | 0.604 | 0.591 |
| 50 | 0.694 | 0.685 | 0.739 | 0.733 | 0.710 | 0.683 |
| 100 | **0.939** | **0.937** | **0.954** | **0.957** | **0.939** | **0.938** |

| Medium relative effect size: $\epsilon = 1.0$ | | | | | |
|---|---|---|---|---|---|
| N | t-test | Welch | Mann-Whit. | r. t-test | boot. | permut. |
| 2 | 0.102 | 0.052 | 0.000 | 0.000 | 0.431 | 0.430 |
| 3 | 0.140 | 0.086 | 0.000 | 0.220 | 0.373 | 0.196 |
| 5 | 0.258 | 0.232 | 0.178 | 0.267 | 0.434 | 0.242 |
| 10 | 0.539 | 0.539 | 0.467 | 0.510 | 0.633 | 0.532 |
| 20 | **0.868** | **0.870** | **0.807** | **0.804** | **0.887** | **0.869** |
| 30 | **0.969** | **0.970** | **0.928** | **0.937** | **0.973** | **0.971** |

| Large relative effect size: $\epsilon = 2.0$ | | | | | |
|---|---|---|---|---|---|
| N | t-test | Welch | Mann-Whit. | r. t-test | boot. | permut. |
| 2 | 0.198 | 0.103 | 0.000 | 0.000 | 0.723 | 0.725 |
| 3 | 0.388 | 0.296 | 0.000 | 0.547 | 0.792 | 0.514 |
| 5 | 0.786 | 0.776 | 0.619 | 0.735 | **0.912** | 0.757 |
| 10 | **0.994** | **0.994** | **0.966** | **0.973** | **0.996** | **0.994** |

Table 4: **Statistical power when comparing samples from two log-normal distribution with equal standard deviation.** The first is centered in $0$ ($\mu_1 = 0$, mean or median depending on the test), the other shifted by the relative effect size ($\mu_2 = \epsilon \, \sigma_{pool}$). Both have same standard deviation $\sigma_1 = \sigma_2 = 1$. Each result represents the percentage of true positive over $10.000$ repetitions. In bold are results satisfying a true positive rate above $0.8$.

| Small relative effect size: $\epsilon = 0.5$ | | | | | |
|---|---|---|---|---|---|
| N | t-test | Welch | Mann-Whit. | r. t-test | boot. | permut. |
| 2 | 0.067 | 0.032 | 0.000 | 0.000 | 0.388 | 0.387 |
| 3 | 0.099 | 0.057 | 0.000 | 0.195 | 0.288 | 0.189 |
| 5 | 0.154 | 0.121 | 0.129 | 0.198 | 0.265 | 0.183 |
| 10 | 0.247 | 0.247 | 0.329 | 0.369 | 0.317 | 0.273 |
| 20 | 0.404 | 0.401 | 0.628 | 0.632 | 0.432 | 0.424 |
| 30 | 0.533 | 0.539 | **0.802** | **0.804** | 0.560 | 0.536 |
| 40 | 0.649 | 0.635 | **0.897** | **0.900** | 0.659 | 0.641 |
| 50 | 0.724 | 0.719 | **0.955** | **0.960** | 0.746 | 0.726 |
| 100 | **0.938** | **0.935** | **1.000** | **1.000** | **0.945** | **0.937** |

| Medium relative effect size: $\epsilon = 1.0$ | | | | | |
|---|---|---|---|---|---|
| N | t-test | Welch | Mann-Whit. | r. t-test | boot. | permut. |
| 2 | 0.147 | 0.070 | 0.000 | 0.000 | 0.609 | 0.603 |
| 3 | 0.262 | 0.193 | 0.000 | 0.428 | 0.542 | 0.412 |
| 5 | 0.431 | 0.397 | 0.379 | 0.458 | 0.584 | 0.475 |
| 10 | 0.657 | 0.649 | 0.768 | 0.796 | 0.726 | 0.671 |
| 20 | **0.876** | **0.864** | **0.979** | **0.978** | **0.902** | **0.876** |
| 30 | **0.953** | **0.954** | **0.999** | **0.998** | **0.964** | **0.954** |

| Large relative effect size: $\epsilon = 2.0$ | | | | | |
|---|---|---|---|---|---|
| N | t-test | Welch | Mann-Whit. | r. t-test | boot. | permut. |
| 2 | 0.357 | 0.191 | 0.000 | 0.000 | **0.858** | **0.860** |
| 3 | 0.642 | 0.534 | 0.000 | 0.769 | **0.858** | 0.744 |
| 5 | **0.838** | **0.812** | 0.738 | **0.801** | **0.916** | **0.843** |
| 10 | **0.960** | **0.959** | **0.985** | **0.988** | **0.979** | **0.964** |

## 8.2 Comparing same distributions with different standard deviations.

Table 5: **Statistical power when comparing samples from two log-normal distribution with different standard deviation.** The first is centered in 0 ($\mu_1 = 0$, $\sigma_1 = 1$ mean or median depending on the test), the other shifted by the relative effect size ($\mu_2 = \epsilon\, \sigma_{pool}$, $\sigma_2 = 2$). Both have same standard deviation $\sigma_1 = \sigma_2 = 1$. Each result represents the percentage of true positive over 10.000 repetitions. In bold are results satisfying a true positive rate above 0.8.

| Small relative effect size: $\epsilon = 0.5$ | | | | | |
|---|---|---|---|---|---|
| N | t-test | Welch | Mann-Whit. | r. t-test | boot. | permut. |
| 2 | 0.062 | 0.030 | 0.000 | 0.000 | 0.310 | 0.314 |
| 3 | 0.084 | 0.058 | 0.000 | 0.152 | 0.244 | 0.138 |
| 5 | 0.110 | 0.097 | 0.079 | 0.115 | 0.217 | 0.120 |
| 10 | 0.191 | 0.180 | 0.174 | 0.195 | 0.261 | 0.189 |
| 20 | 0.342 | 0.333 | 0.329 | 0.315 | 0.385 | 0.330 |
| 30 | 0.476 | 0.477 | 0.454 | 0.466 | 0.509 | 0.479 |
| 40 | 0.602 | 0.585 | 0.574 | 0.573 | 0.622 | 0.598 |
| 50 | 0.690 | 0.696 | 0.664 | 0.674 | 0.713 | 0.693 |
| 100 | **0.937** | **0.941** | **0.923** | **0.924** | **0.945** | **0.938** |

| Medium relative effect size: $\epsilon = 1.0$ | | | | | |
|---|---|---|---|---|---|
| N | t-test | Welch | Mann-Whit. | r. t-test | boot. | permut. |
| 2 | 0.112 | 0.053 | 0.000 | 0.000 | 0.484 | 0.476 |
| 3 | 0.171 | 0.137 | 0.000 | 0.284 | 0.428 | 0.274 |
| 5 | 0.303 | 0.253 | 0.228 | 0.280 | 0.466 | 0.313 |
| 10 | 0.572 | 0.531 | 0.513 | 0.542 | 0.651 | 0.572 |
| 20 | **0.864** | **0.861** | **0.837** | **0.839** | **0.891** | **0.858** |
| 30 | **0.968** | **0.961** | **0.949** | **0.955** | **0.971** | **0.965** |

| Large relative effect size: $\epsilon = 2.0$ | | | | | |
|---|---|---|---|---|---|
| N | t-test | Welch | Mann-Whit. | r. t-test | boot. | permut. |
| 2 | 0.248 | 0.130 | 0.000 | 0.000 | 0.783 | 0.787 |
| 3 | 0.479 | 0.371 | 0.000 | 0.655 | **0.808** | 0.621 |
| 5 | 0.785 | 0.734 | 0.676 | 0.756 | **0.915** | 0.800 |
| 10 | **0.988** | **0.983** | **0.973** | **0.979** | **0.994** | **0.987** |

Table 6: **Statistical power when comparing samples from two bimodal distribution with different standard deviation.** The first is centered in $0$ ($\mu_1 = 0$, $\sigma_1 = 1$ mean or median depending on the test), the other shifted by the relative effect size ($\mu_2 = \epsilon\,\sigma_{pool}$, $\sigma_2 = 2$). Both have same standard deviation $\sigma_1 = \sigma_2 = 1$. Each result represents the percentage of true positive over 10.000 repetitions. In bold are results satisfying a true positive rate above 0.8.

| Small relative effect size: $\epsilon = 0.5$ | | | | | |
| N | t-test | Welch | Mann-Whit. | r. t-test | boot. | permut. |
| --- | --- | --- | --- | --- | --- | --- |
| 2 | 0.115 | 0.067 | 0.000 | 0.000 | 0.283 | 0.280 |
| 3 | 0.117 | 0.088 | 0.000 | 0.132 | 0.206 | 0.126 |
| 5 | 0.092 | 0.080 | 0.044 | 0.071 | 0.205 | 0.088 |
| 10 | 0.168 | 0.159 | 0.094 | 0.112 | 0.241 | 0.163 |
| 20 | 0.312 | 0.310 | 0.174 | 0.169 | 0.370 | 0.318 |
| 30 | 0.472 | 0.440 | 0.218 | 0.225 | 0.496 | 0.455 |
| 40 | 0.587 | 0.587 | 0.266 | 0.277 | 0.615 | 0.590 |
| 50 | 0.685 | 0.690 | 0.331 | 0.322 | 0.708 | 0.688 |
| 100 | **0.943** | **0.941** | 0.551 | 0.548 | **0.941** | **0.943** |

| Medium relative effect size: $\epsilon = 1.0$ | | | | | |
| N | t-test | Welch | Mann-Whit. | r. t-test | boot. | permut. |
| --- | --- | --- | --- | --- | --- | --- |
| 2 | 0.164 | 0.096 | 0.000 | 0.000 | 0.374 | 0.380 |
| 3 | 0.137 | 0.112 | 0.000 | 0.182 | 0.371 | 0.175 |
| 5 | 0.229 | 0.188 | 0.121 | 0.197 | 0.411 | 0.212 |
| 10 | 0.528 | 0.507 | 0.325 | 0.366 | 0.621 | 0.509 |
| 20 | **0.871** | **0.857** | 0.631 | 0.624 | **0.896** | **0.872** |
| 30 | **0.972** | **0.974** | 0.797 | **0.805** | **0.974** | **0.971** |
| 40 | **0.995** | **0.993** | **0.897** | **0.896** | **0.995** | **0.995** |
| 50 | **0.999** | **0.999** | **0.954** | **0.954** | **0.999** | **0.999** |

| Large relative effect size: $\epsilon = 2.0$ | | | | | |
| N | t-test | Welch | Mann-Whit. | r. t-test | boot. | permut. |
| --- | --- | --- | --- | --- | --- | --- |
| 2 | 0.267 | 0.162 | 0.000 | 0.000 | 0.768 | 0.764 |
| 3 | 0.319 | 0.189 | 0.000 | 0.600 | 0.770 | 0.571 |
| 5 | 0.787 | 0.722 | 0.690 | 0.800 | **0.923** | **0.816** |
| 10 | **0.997** | **0.997** | **0.981** | **0.987** | **0.999** | **0.998** |

Table 7: **Statistical power when comparing samples from two log-normal distribution with different standard deviation.** The first is centered in 0 ($\mu_1 = 0$, $\sigma_1 = 1$ mean or median depending on the test), the other shifted by the relative effect size ($\mu_2 = \epsilon\, \sigma_{pool}$, $\sigma_2 = 2$). Both have same standard deviation $\sigma_1 = \sigma_2 = 1$. Each result represents the percentage of true positive over 10.000 repetitions. In bold are results satisfying a true positive rate above 0.8.

| N | t-test | Welch | Mann-Whit. | r. t-test | boot. | permut. |
|---|--------|-------|------------|-----------|-------|---------|
| Small relative effect size: $\epsilon$=0.5 | | | | | | |
| 2 | 0.057 | 0.030 | 0.000 | 0.000 | 0.408 | 0.408 |
| 3 | 0.056 | 0.038 | 0.000 | 0.299 | 0.242 | 0.157 |
| 5 | 0.092 | 0.063 | 0.246 | 0.342 | 0.199 | 0.145 |
| 10 | 0.185 | 0.168 | 0.588 | 0.612 | 0.233 | 0.249 |
| 20 | 0.394 | 0.379 | **0.898** | **0.901** | 0.389 | 0.445 |
| 30 | 0.571 | 0.560 | **0.980** | **0.983** | 0.549 | 0.607 |
| 40 | 0.693 | 0.693 | **0.997** | **0.997** | 0.674 | 0.732 |
| 50 | **0.802** | 0.800 | **0.999** | **0.999** | 0.781 | **0.816** |
| 100 | **0.980** | **0.978** | **1.000** | **1.000** | **0.974** | **0.983** |
| Small relative effect size: $\epsilon$=1.0 | | | | | | |
| 2 | 0.147 | 0.072 | 0.000 | 0.000 | 0.680 | 0.685 |
| 3 | 0.264 | 0.181 | 0.000 | 0.631 | 0.581 | 0.474 |
| 5 | 0.464 | 0.401 | 0.604 | 0.707 | 0.637 | 0.567 |
| 10 | 0.749 | 0.728 | **0.956** | **0.966** | 0.787 | **0.801** |
| 20 | **0.951** | **0.945** | **1.000** | **1.000** | **0.951** | **0.964** |
| Small relative effect size: $\epsilon$=2.0 | | | | | | |
| 2 | 0.419 | 0.230 | 0.000 | 0.000 | **0.924** | **0.930** |
| 3 | 0.722 | 0.596 | 0.000 | **0.911** | **0.921** | **0.842** |
| 5 | **0.904** | **0.868** | **0.910** | **0.935** | **0.959** | **0.944** |
| 10 | **0.989** | **0.986** | **0.999** | **1.000** | **0.993** | **0.994** |

## 8.3 Comparing different distributions with equal standard deviations.

Table 8: **Statistical power when comparing samples from a normal distribution and a log-normal distribution with equal standard deviation.** The first is centered in $0$ ($\mu_1 = 0$, $\sigma_1 = 1$ mean or median depending on the test), the other shifted by the relative effect size ($\mu_2 = \epsilon\,\sigma_{pool}$, $\sigma_2 = 1$). Each result represents the percentage of true positive over $10.000$ repetitions. In bold are results satisfying a true positive rate above $0.8$.

| N | t-test | Welch | Mann-Whit. | r. t-test | boot. | permut. |
|---|--------|-------|-----------|-----------|-------|---------|
| Small relative effect size: $\epsilon = 0.5$ | | | | | | |
| 2 | 0.040 | 0.016 | 0.000 | 0.000 | 0.270 | 0.267 |
| 3 | 0.047 | 0.032 | 0.000 | 0.181 | 0.187 | 0.099 |
| 5 | 0.076 | 0.062 | 0.125 | 0.189 | 0.164 | 0.088 |
| 10 | 0.155 | 0.145 | 0.330 | 0.357 | 0.211 | 0.169 |
| 20 | 0.315 | 0.320 | 0.628 | 0.621 | 0.352 | 0.335 |
| 30 | 0.483 | 0.484 | 0.797 | 0.792 | 0.505 | 0.494 |
| 40 | 0.611 | 0.615 | **0.894** | **0.903** | 0.620 | 0.619 |
| 50 | 0.729 | 0.725 | **0.951** | **0.953** | 0.731 | 0.725 |
| 100 | **0.958** | **0.959** | **0.999** | **0.999** | **0.959** | **0.960** |
| Medium relative effect size: $\epsilon = 1.0$ | | | | | | |
| 2 | 0.078 | 0.042 | 0.000 | 0.000 | 0.477 | 0.472 |
| 3 | 0.137 | 0.098 | 0.000 | 0.390 | 0.411 | 0.263 |
| 5 | 0.277 | 0.251 | 0.368 | 0.480 | 0.457 | 0.307 |
| 10 | 0.600 | 0.584 | 0.785 | **0.805** | 0.666 | 0.613 |
| 20 | **0.912** | **0.910** | **0.981** | **0.979** | **0.924** | **0.919** |
| Large relative effect size: $\epsilon = 2.0$ | | | | | | |
| 2 | 0.230 | 0.113 | 0.000 | 0.000 | **0.856** | **0.841** |
| 3 | 0.513 | 0.392 | 0.000 | **0.838** | **0.864** | 0.709 |
| 5 | **0.863** | **0.815** | **0.872** | **0.927** | **0.946** | **0.880** |
| 10 | **0.997** | **0.997** | **0.999** | **0.999** | **0.999** | **0.999** |

Table 9: **Statistical power when comparing samples from a log-normal distribution and a bimodal distribution with equal standard deviation.** The first is centered in 0 ($\mu_1 = 0$, $\sigma_1 = 1$ mean or median depending on the test), the other shifted by the relative effect size ($\mu_2 = \epsilon\,\sigma_{pool}$, $\sigma_2 = 1$). Each result represents the percentage of true positive over 10.000 repetitions. In bold are results satisfying a true positive rate above 0.8.

| N | t-test | Welch | Mann-Whit. | r. t-test | boot. | permut. |
|---|--------|-------|------------|-----------|-------|---------|
| Small relative effect size: $\epsilon = 0.5$ | | | | | | |
| 2 | 0.100 | 0.055 | 0.000 | 0.000 | 0.337 | 0.330 |
| 3 | 0.101 | 0.078 | 0.000 | 0.102 | 0.276 | 0.143 |
| 5 | 0.149 | 0.121 | 0.045 | 0.063 | 0.262 | 0.137 |
| 10 | 0.239 | 0.228 | 0.074 | 0.084 | 0.318 | 0.238 |
| 20 | 0.390 | 0.395 | 0.125 | 0.128 | 0.447 | 0.381 |
| 30 | 0.502 | 0.510 | 0.167 | 0.172 | 0.561 | 0.509 |
| 40 | 0.611 | 0.603 | 0.201 | 0.213 | 0.641 | 0.604 |
| 50 | 0.691 | 0.686 | 0.250 | 0.242 | 0.725 | 0.693 |
| 100 | **0.920** | **0.917** | 0.427 | 0.429 | **0.929** | **0.922** |
| Medium relative effect size: $\epsilon = 1.0$ | | | | | | |
| 2 | 0.161 | 0.085 | 0.000 | 0.000 | 0.513 | 0.507 |
| 3 | 0.202 | 0.136 | 0.000 | 0.221 | 0.484 | 0.294 |
| 5 | 0.358 | 0.316 | 0.161 | 0.225 | 0.534 | 0.351 |
| 10 | 0.601 | 0.606 | 0.374 | 0.413 | 0.694 | 0.600 |
| 20 | **0.839** | **0.843** | 0.699 | 0.707 | **0.881** | **0.846** |
| 30 | **0.939** | **0.940** | **0.865** | **0.869** | **0.954** | **0.940** |
| 40 | **0.978** | **0.980** | **0.947** | **0.950** | **0.983** | **0.980** |
| Large relative effect size: $\epsilon = 2.0$ | | | | | | |
| 2 | 0.275 | 0.158 | 0.000 | 0.000 | **0.808** | **0.809** |
| 3 | 0.539 | 0.390 | 0.000 | 0.585 | **0.819** | 0.647 |
| 5 | **0.804** | 0.781 | 0.613 | 0.719 | **0.898** | 0.792 |
| 10 | **0.955** | **0.953** | **0.956** | **0.962** | **0.976** | **0.956** |

Table 10: **Statistical power when comparing samples from a normal distribution and a bimodal distribution with equal standard deviation.** The first is centered in 0 ($\mu_1 = 0$, $\sigma_1 = 1$ mean or median depending on the test), the other shifted by the relative effect size ($\mu_2 = \epsilon\,\sigma_{pool}$, $\sigma_2 = 1$). Each result represents the percentage of true positive over 10.000 repetitions. In bold are results satisfying a true positive rate above 0.8.

| N | t-test | Welch | Mann-Whit. | r. t-test | boot. | permut. |
|---|---|---|---|---|---|---|
| Small relative effect size: $\epsilon = 0.5$ | | | | | | |
| 2 | 0.061 | 0.031 | 0.000 | 0.000 | 0.287 | 0.282 |
| 3 | 0.070 | 0.051 | 0.000 | 0.112 | 0.217 | 0.102 |
| 5 | 0.104 | 0.089 | 0.061 | 0.101 | 0.200 | 0.093 |
| 10 | 0.175 | 0.173 | 0.141 | 0.160 | 0.246 | 0.173 |
| 20 | 0.334 | 0.339 | 0.281 | 0.283 | 0.384 | 0.326 |
| 30 | 0.466 | 0.472 | 0.410 | 0.411 | 0.515 | 0.474 |
| 40 | 0.596 | 0.590 | 0.513 | 0.519 | 0.607 | 0.582 |
| 50 | 0.695 | 0.683 | 0.614 | 0.611 | 0.709 | 0.693 |
| 100 | **0.938** | **0.938** | **0.887** | **0.887** | **0.940** | **0.937** |
| Medium relative effect size: $\epsilon = 1.0$ | | | | | | |
| 2 | 0.107 | 0.051 | 0.000 | 0.000 | 0.430 | 0.425 |
| 3 | 0.145 | 0.099 | 0.000 | 0.230 | 0.406 | 0.212 |
| 5 | 0.261 | 0.244 | 0.180 | 0.266 | 0.451 | 0.256 |
| 10 | 0.550 | 0.545 | 0.463 | 0.502 | 0.647 | 0.545 |
| 20 | **0.866** | **0.867** | **0.811** | **0.808** | **0.889** | **0.869** |
| 30 | **0.968** | **0.971** | **0.937** | **0.934** | **0.975** | **0.967** |
| Large relative effect size: $\epsilon = 2.0$ | | | | | | |
| 2 | 0.198 | 0.105 | 0.000 | 0.000 | 0.763 | 0.768 |
| 3 | 0.427 | 0.323 | 0.000 | 0.596 | 0.797 | 0.569 |
| 5 | 0.794 | 0.763 | 0.667 | 0.783 | **0.914** | 0.783 |
| 10 | **0.991** | **0.989** | **0.979** | **0.983** | **0.996** | **0.990** |

## 8.4 Comparing different distributions with different standard deviations.

Table 11: **Statistical power when comparing samples from a log-normal distribution and a bimodal distribution with different standard deviation.** The first is centered in $0$ ($\mu_1 = 0$, $\sigma_1 = 1$ mean or median depending on the test), the other shifted by the relative effect size ($\mu_2 = \epsilon \, \sigma_{pool}$, $\sigma_2 = 2$). Each result represents the percentage of true positive over 10.000 repetitions. In bold are results satisfying a true positive rate above $0.8$.

| N | t-test | Welch | Mann-Whit. | r. t-test | boot. | permut. |
|---|--------|-------|------------|-----------|-------|---------|
| \multicolumn{7}{l}{Small relative effect size: $\epsilon = 0.5$} | | | | | | |
| 2 | 0.162 | 0.111 | 0.000 | 0.000 | 0.257 | 0.248 |
| 3 | 0.105 | 0.097 | 0.000 | 0.115 | 0.256 | 0.109 |
| 5 | 0.111 | 0.085 | 0.030 | 0.036 | 0.205 | 0.113 |
| 10 | 0.182 | 0.167 | 0.044 | 0.048 | 0.265 | 0.179 |
| 20 | 0.339 | 0.324 | 0.045 | 0.047 | 0.389 | 0.331 |
| 30 | 0.470 | 0.475 | 0.051 | 0.051 | 0.512 | 0.477 |
| 40 | 0.588 | 0.592 | 0.059 | 0.056 | 0.620 | 0.595 |
| 50 | 0.698 | 0.691 | 0.057 | 0.064 | 0.712 | 0.686 |
| 100 | **0.939** | **0.936** | 0.089 | 0.092 | **0.938** | **0.934** |
| \multicolumn{7}{l}{Medium relative effect size: $\epsilon = 1.0$} | | | | | | |
| 2 | 0.200 | 0.129 | 0.000 | 0.000 | 0.387 | 0.388 |
| 3 | 0.126 | 0.110 | 0.000 | 0.155 | 0.408 | 0.189 |
| 5 | 0.240 | 0.182 | 0.076 | 0.113 | 0.439 | 0.227 |
| 10 | 0.541 | 0.521 | 0.160 | 0.181 | 0.636 | 0.536 |
| 20 | **0.870** | **0.852** | 0.301 | 0.304 | **0.886** | **0.857** |
| 30 | **0.964** | **0.962** | 0.424 | 0.417 | **0.969** | **0.960** |
| 40 | **0.992** | **0.989** | 0.520 | 0.526 | **0.992** | **0.991** |
| 50 | **0.998** | **0.998** | 0.613 | 0.618 | **0.998** | **0.998** |
| 100 | **1.000** | **1.000** | **0.883** | **0.885** | **1.000** | **1.000** |
| \multicolumn{7}{l}{Large relative effect size: $\epsilon = 2.0$} | | | | | | |
| 2 | 0.294 | 0.209 | 0.000 | 0.000 | **0.830** | **0.837** |
| 3 | 0.393 | 0.197 | 0.000 | 0.636 | 0.787 | 0.685 |
| 5 | **0.806** | 0.739 | 0.705 | **0.802** | **0.918** | 0.845 |
| 10 | **0.984** | **0.984** | **0.985** | **0.988** | **0.991** | **0.985** |

Table 12: **Statistical power when comparing samples from a normal distribution and a bimodal distribution with different standard deviation.** The first is centered in 0 ($\mu_1 = 0$, $\sigma_1 = 1$ mean or median depending on the test), the other shifted by the relative effect size ($\mu_2 = \epsilon \, \sigma_{pool}$, $\sigma_2 = 2$). Each result represents the percentage of true positive over 10.000 repetitions. In bold are results satisfying a true positive rate above 0.8.

| N | t-test | Welch | Mann-Whit. | r. t-test | boot. | permut. |
|---|--------|-------|------------|-----------|-------|---------|
| Small relative effect size: $\epsilon = 0.5$ | | | | | | |
| 2 | 0.130 | 0.075 | 0.000 | 0.000 | 0.270 | 0.263 |
| 3 | 0.109 | 0.089 | 0.000 | 0.128 | 0.219 | 0.115 |
| 5 | 0.096 | 0.080 | 0.038 | 0.061 | 0.200 | 0.100 |
| 10 | 0.165 | 0.157 | 0.084 | 0.097 | 0.245 | 0.164 |
| 20 | 0.324 | 0.311 | 0.132 | 0.134 | 0.374 | 0.316 |
| 30 | 0.460 | 0.466 | 0.172 | 0.180 | 0.497 | 0.469 |
| 40 | 0.584 | 0.591 | 0.214 | 0.209 | 0.616 | 0.594 |
| 50 | 0.699 | 0.693 | 0.244 | 0.252 | 0.707 | 0.688 |
| 100 | **0.942** | **0.941** | 0.409 | 0.408 | **0.942** | **0.940** |
| Medium relative effect size: $\epsilon = 1.0$ | | | | | | |
| 2 | 0.173 | 0.100 | 0.000 | 0.000 | 0.353 | 0.356 |
| 3 | 0.131 | 0.113 | 0.000 | 0.173 | 0.384 | 0.166 |
| 5 | 0.225 | 0.189 | 0.099 | 0.167 | 0.414 | 0.209 |
| 10 | 0.526 | 0.508 | 0.267 | 0.303 | 0.621 | 0.518 |
| 20 | **0.881** | **0.862** | 0.529 | 0.524 | **0.891** | **0.867** |
| 30 | **0.972** | **0.972** | 0.706 | 0.704 | **0.974** | **0.970** |
| 40 | **0.995** | **0.995** | **0.826** | **0.822** | **0.995** | **0.995** |
| 50 | **0.999** | **0.999** | **0.894** | **0.896** | **1.000** | **0.999** |
| 100 | **1.000** | **1.000** | **0.996** | **0.994** | **1.000** | **1.000** |
| Large relative effect size: $\epsilon = 2.0$ | | | | | | |
| 2 | 0.265 | 0.160 | 0.000 | 0.000 | 0.789 | 0.798 |
| 3 | 0.341 | 0.200 | 0.000 | 0.648 | 0.771 | 0.624 |
| 5 | 0.795 | 0.728 | 0.740 | **0.836** | **0.927** | **0.836** |
| 10 | **0.997** | **0.997** | **0.991** | **0.993** | **0.999** | **0.997** |

Table 13: **Statistical power when comparing samples from a normal distribution and a log-normal distribution with different standard deviation.** The first is centered in 0 ($\mu_1 = 0$, $\sigma_1 = 1$ mean or median depending on the test), the other shifted by the relative effect size ($\mu_2 = \epsilon \, \sigma_{pool}$, $\sigma_2 = 2$). Each result represents the percentage of true positive over 10.000 repetitions. In bold are results satisfying a true positive rate above 0.8.

Small relative effect size: $\epsilon = 0.5$

| N | t-test | Welch | Mann-Whit. | r. t-test | boot. | permut. |
|---|--------|-------|-----------|-----------|-------|---------|
| 2 | 0.034 | 0.012 | 0.000 | 0.000 | 0.263 | 0.261 |
| 3 | 0.033 | 0.023 | 0.000 | 0.277 | 0.163 | 0.090 |
| 5 | 0.053 | 0.043 | 0.233 | 0.325 | 0.139 | 0.081 |
| 10 | 0.135 | 0.128 | 0.581 | 0.618 | 0.180 | 0.173 |
| 20 | 0.337 | 0.314 | **0.899** | **0.901** | 0.349 | 0.394 |
| 30 | 0.532 | 0.537 | **0.980** | **0.975** | 0.508 | 0.586 |
| 40 | 0.699 | 0.690 | **0.996** | **0.996** | 0.666 | 0.725 |
| 50 | **0.804** | **0.805** | **0.999** | **0.999** | 0.772 | **0.823** |
| 100 | **0.986** | **0.986** | **1.000** | **1.000** | **0.982** | **0.989** |

Medium relative effect size: $\epsilon = 1.0$

| N | t-test | Welch | Mann-Whit. | r. t-test | boot. | permut. |
|---|--------|-------|-----------|-----------|-------|---------|
| 2 | 0.088 | 0.039 | 0.000 | 0.000 | 0.554 | 0.558 |
| 3 | 0.151 | 0.108 | 0.000 | 0.637 | 0.474 | 0.340 |
| 5 | 0.345 | 0.301 | 0.671 | 0.781 | 0.560 | 0.447 |
| 10 | 0.748 | 0.722 | **0.977** | **0.981** | 0.794 | **0.817** |
| 20 | **0.977** | **0.973** | **1.000** | **1.000** | **0.966** | **0.989** |

Large relative effect size: $\epsilon = 2.0$

| N | t-test | Welch | Mann-Whit. | r. t-test | boot. | permut. |
|---|--------|-------|-----------|-----------|-------|---------|
| 2 | 0.323 | 0.161 | 0.000 | 0.000 | **0.962** | **0.965** |
| 3 | 0.700 | 0.553 | 0.000 | **0.985** | **0.944** | **0.887** |
| 5 | **0.952** | **0.917** | **0.992** | **0.997** | **0.977** | **0.992** |

Table 14: **Statistical power when comparing samples from empirical RL distribution** (192*samples*) **from the SAC algorithm and the TD3 algorithm, both run in the Half-Cheetah environment for 2M steps.** The first is centered in 0 ($\mu_1 = 0$, $\sigma_1$=1313 mean or median depending on the test), the other shifted by the relative effect size ($\mu_2 = \epsilon\,\sigma_{pool}$, $\sigma_2$=1508). Each result represents the percentage of true positive over 10.000 repetitions. In bold are results satisfying a true positive rate above 0.8.

| N | t-test | Welch | Mann-Whit. | r. t-test | boot. | permut. |
|---|---|---|---|---|---|---|
| Small relative effect size: $\epsilon = 0.5$ | | | | | | |
| 2 | 0.113 | 0.052 | 0.000 | 0.000 | 0.427 | 0.431 |
| 3 | 0.157 | 0.133 | 0.000 | 0.216 | 0.330 | 0.258 |
| 5 | 0.181 | 0.162 | 0.119 | 0.154 | 0.287 | 0.208 |
| 10 | 0.248 | 0.230 | 0.263 | 0.293 | 0.307 | 0.257 |
| 20 | 0.380 | 0.374 | 0.486 | 0.489 | 0.417 | 0.400 |
| 30 | 0.504 | 0.506 | 0.655 | 0.665 | 0.529 | 0.523 |
| 40 | 0.629 | 0.621 | 0.776 | 0.776 | 0.647 | 0.637 |
| 50 | 0.716 | 0.722 | **0.858** | **0.861** | 0.735 | 0.722 |
| 100 | **0.951** | **0.951** | **0.990** | **0.988** | **0.953** | **0.950** |
| Medium relative effect size: $\epsilon = 1.0$ | | | | | | |
| 2 | 0.210 | 0.106 | 0.000 | 0.000 | 0.588 | 0.591 |
| 3 | 0.296 | 0.248 | 0.000 | 0.380 | 0.534 | 0.424 |
| 5 | 0.403 | 0.360 | 0.265 | 0.339 | 0.565 | 0.424 |
| 10 | 0.622 | 0.592 | 0.591 | 0.618 | 0.705 | 0.640 |
| 20 | **0.883** | **0.876** | **0.887** | **0.888** | **0.901** | **0.884** |
| 30 | **0.970** | **0.968** | **0.973** | **0.973** | **0.976** | **0.972** |
| Large relative effect size: $\epsilon = 2.0$ | | | | | | |
| 2 | 0.411 | 0.227 | 0.000 | 0.000 | **0.833** | **0.837** |
| 3 | 0.588 | 0.474 | 0.000 | 0.697 | **0.839** | 0.737 |
| 5 | **0.803** | 0.751 | 0.648 | 0.708 | **0.910** | **0.823** |
| 10 | **0.983** | **0.979** | **0.960** | **0.967** | **0.991** | **0.981** |

## 8.5 Comparison of two empirical distributions with unequal variance, SAC and TD3

Table 15: **Statistical power when comparing SAC and TD3** with relative effect sizes $\epsilon = 0.80$ for median tests (Mann-Whitney, Ranked t-test), $\epsilon = 0.93$ for other tests. The true relative effect sizes are estimated using 192 samples of each distribution.

| N | t-test | Welch | Mann-Whit. | r. t-test | boot. | permut. |
|---|---|---|---|---|---|---|
| 2 | 0.000 | 0.000 | 0.113 | 0.059 | 0.604 | 0.596 |
| 3 | 0.000 | 0.410 | 0.196 | 0.125 | 0.500 | 0.380 |
| 5 | 0.379 | 0.475 | 0.388 | 0.304 | 0.570 | 0.475 |
| 7 | 0.571 | 0.629 | 0.522 | 0.482 | 0.632 | 0.575 |
| 10 | 0.767 | 0.793 | 0.664 | 0.638 | 0.705 | 0.670 |
| 15 | **0.927** | **0.933** | 0.780 | 0.778 | **0.809** | 0.782 |
| 20 | **0.981** | **0.983** | **0.837** | **0.842** | **0.862** | **0.835** |
| 30 | **0.999** | **0.999** | **0.892** | **0.891** | **0.930** | **0.907** |
| 40 | **1.000** | **1.000** | **0.950** | **0.951** | **0.966** | **0.953** |

