# OpenReview forum: "A Hitchhiker's Guide to Statistical Comparisons of Reinforcement Learning Algorithms"
_ICLR.cc/2019/Workshop/RML — RML 2019_

### Official Review · AnonReviewer1 · 2019-04-01
**Interesting work, but relevance to RL is somewhat uncertain**

**Rating:** 3
**Confidence:** 3

**Review:**

Summary: This paper has an interesting investigation of significance testing for RL, however I am skeptical about whether significance testing is appropriate for comparing RL experiments.

Notes:
  -Abstract asserts that checking statistical significance is the first step towards reproducibility, which seems somewhat debatable to me.
  -Paper should probably just say that it’s about statistical significance in the title.

Comments:
  -This is a matter of judgement, but I feel like statistical significance is useful for cases where there is a tiny effect size, and we want to make sure that it isn’t just random noise.  For example, one of the earliest uses for statistical significance was for studying the sex difference observed in the number of newborn babies (like 50.00001% are male).  In this case it’s important to know if this result is statistically significant or just random noise.  In the case of different RL algorithms, I’d be hard-pressed to imagine a situation where we’d want to publish a new algorithm where the improvement isn’t obviously statistically significant, especially because the power of the statistical test can easily be improved just by running more trials.
  -I can think of one exception to this, which is if we ran our RL algorithm in the real-world, and then we had a breakdown of many different evaluation categories, and then we wanted to identify which categories had significant or non-significant improvements.
  -One useful outcome of this is determining the sample sizes needed for certain levels of significance, which could be helpful for determining how many trials to run for RL experiments.
  -Some interesting recommendations are also made regarding the choice of significance tests.

---

### Decision · Program_Chairs · 2019-04-05
**Acceptance Decision**

Accept